# Effective and Sustained Control of Soil-Borne Plant Diseases by Biodegradable Polyhydroxybutyrate Mulch Films Embedded with Fungicide of Prothioconazole

**DOI:** 10.3390/molecules26030762

**Published:** 2021-02-02

**Authors:** Ge Chen, Lidong Cao, Chong Cao, Pengyue Zhao, Fengmin Li, Bo Xu, Qiliang Huang

**Affiliations:** 1Institute of Plant Protection, Chinese Academy of Agricultural Sciences, No. 2 Yuanmingyuan West Road, Haidian District, Beijing 100193, China; chenge0036@126.com (G.C.); ccao@ippcaas.cn (C.C.); zhaopengyue@caas.cn (P.Z.); fmli@ippcaas.cn (F.L.); 2Henan Haonianjing Biological Development Co., Ltd., Yangjin Industrial Park, Jinshui District, Zhengzhou 450000, China; haonianjing01@163.com

**Keywords:** antifungal mulch film, prothioconazole, polyhydroxybutyrate, controlled release, soilborne plant diseases

## Abstract

Soil-borne diseases and plant rhizosphere nematode have caused many crop yield losses. Increased environmental awareness is leading to more restrictions on the use of certain fumigants and root irrigation methods due to their impact on human health and soil system. Therefore, it is necessary to find alternative treatments to maintain crop economic yields and environmental sustainability. In the present work, biodegradable antifungal mulches were prepared by blending poly(3-hydroxybutyrate-co-4-hydroxybutyrate) (PHB) with fungicide of prothioconazole (PRO), which were used for effective and sustained control of soil-borne plant diseases. To reveal the application prospect of the PHB/PRO composite films in the management of soilborne plant diseases, some physical and biological properties were evaluated. The proper mulch film of PHB/PRO was assessed based on its mechanical and optical properties, while water solubility and the film micromorphology was further characterized. The release patterns of composite films under different pH levels were investigated. Moreover, the in vitro antifungal bioassay and pot experiment showed satisfactory bioactivity of the PHB/PRO films against *Sclerotium rolfsii Sacc.*, a soil-borne disease in peanut fields. This study demonstrated that the biodegradable mulch films containing PRO fungicide are capable of inhibiting soil-borne plant pathogenic fungi effectively, and this facile but powerful strategy may find wide applicability in sustainable plant and horticulture protection.

## 1. Introduction

With the rapid development of protected agriculture and intensive cultivation, the occurrence of soil-borne diseases and nematodes is becoming more and more serious. Devastating soil-borne diseases caused by *Fusarium* spp., *Phytophthora* spp., *Verticillium* spp., *Sclerotinia* spp., *Rhizoctonia* spp., *Pythium* spp., and *Meloidogyne* spp. increase year by year [1]. Once soil-borne diseases occur, the therapeutic effect of the agriculture becomes limited. The main treatment is root-irrigation of pesticides or root spray. Although these methods can control the soil-borne diseases efficiently, a sizable quantity of pesticides would leach into the soil, which inevitably causes serious impacts on the soil and groundwater. The application of organic amendments followed by soil plastic mulching can be a good option for the control of soil-borne disease [2]. Other physical or soil fumigation techniques also commonly use plastic films [3,4]. These agricultural plastic mulch films are typically composed of polyethylene [5] and other non-biodegradable polymer materials, which accumulate for a long time in the soil, affect the development of plant roots, hinder soil gas exchange and water infiltration, and change the structure of the soil microbial community [6,7]. Thus, it is highly desirable and promising to use biodegradable antimicrobial mulch films embedded with fungicides to manage soil-borne diseases. Biodegradable antifungal mulch films can prevent and control soil-borne diseases as they degrade in soil to release active ingredients.

Nowadays, a variety of biodegradable polymers have been used to prepare agricultural materials [8]. The end products of the degradation of the polymer materials can be used as an energy and nutritional source for the organisms. Polyhydroxyalkanoates (PHAs) are an environmentally friendly biodegradable material. PHAs degrade either under anaerobic conditions to produce carbon dioxide (CO_2_), water (H_2_O), and methane or under aerobic conditions to produce CO_2_ and H_2_O [9]. Among the plethora of PHAs, polyhydroxybutyrates mainly including homopolymer of poly(3-hydroxybutyrate) (P3HB) and copolymer poly(3-hydroxybutyrate-co-4-hydroxybutyrate) (PHB) have often been used as bases for the construction of controlled release preparations (CRPs) of pesticides, such as particles, microcapsules, microspheres, pellets, and films [10,11,12,13,14,15]. All these preparations can function in the soil for a long time, ensuring gradual and sustained delivery of pesticides. The delivery of pesticides by CRPs is favorable because the release of active ingredients can be regulated well to accommodate the requirements of agronomic practices. Therefore, the main goal of this work is to propose novel, environmental-friendly mulch films embedded with fungicides, which can not only control soil-borne plant diseases effectively, but also offer ecological and economic advantages, alleviating the environmental pressure brought about by non-biodegradable agricultural plastic films and pesticides. In the existing application of PHB-supported pesticide films, the research on the mechanical and optical properties of the films is rarely reported. Other biodegradable materials that integrated herbicides with mulching film for sustained and effective weed control have been reported [16,17]. In the application of biodegradable plastic mulching film for controlling soil-borne disease, Liang et al. prepared biodegradable water-triggered chitosan/hydroxypropylmethyl cellulose metalaxyl mulch film for sustained control of *Phytophthora sojae* in soybean [18].

In the present study, we chose peanut, *Sclerotium rolfsii Sacc.*, prothioconazole (PRO) as the model crop, pathogen, and fungicide, respectively, to demonstrate our promising proposal. *S. rolfsii* is a destructive soil-borne fungal pathogen with a wide host range, which affects crop productivity and causes large scale economic losses. Among its many host genera, peanut sustains high yield losses [19]. Prothioconazole is a highly effective broad-spectrum triazole fungicide, which was developed by Bayer Crop Science and listed in 2004. It can prevent and cure various diseases by inhibiting the C-14α-demethylase, which is involved in the biosynthesis of fungal sterols [20,21]. PRO is rapidly metabolized to prothioconazole-desthio via desulfurization, which mainly occurs in soil, plants, and animals [22]. The slow and sustained release performance of PRO endowed by PHB as carriers has great potential to make up for the disadvantage of PRO in practical application.

Encouraged by this idea, five kinds of PRO-loaded PHB films (PHB/PRO) with different contents were facilely prepared by mixing PRO and PHB in different proportions. The optimal mulch film was selected by evaluating the mechanical and optical properties, and the water solubility. The microstructures of the mulch films were characterized by scanning electron microscope (SEM). Furthermore, the controlled-release ability of PRO and contact angle of water on the mulch film surface were evaluated. In order to explore the application prospect of the mulch films for the prevention of soil-borne plant diseases, the in vitro fungicidal activity and pot experiments were also investigated.

## 2. Results and Discussion

### 2.1. Film Characterization

#### 2.1.1. Film Thickness

The pesticide loading of PHB/PRO composite films prepared with different mass ratios of PHB and PRO was 1.73%, 3.96%, 8.75%, 18.15%, and 37.32%, respectively. The detected content of the films was a little lower than the theoretical content, which is possibly ascribed to the fact that PRO sticks to the container wall due to the high concentration and viscosity of the PHB/PRO blend melted during the preparation process. The thickness of the PHB/PRO composite films with different drug loading are shown in Figure 1. The results obtained showed that the thickness of the films range from 46.6 to 53.8 μm and there is no significant difference between different films.

#### 2.1.2. Mechanical Properties

Figure 2A shows the stress-strain curves of PRO/PHB films with different mass ratios, and it can be intuitively seen that the film changes in the process of stretching. This can also be seen from Figure 2B,C, when the PRO content is 8.75%, the tensile strength and elongation at break reach the maximum, and then the corresponding values decrease with the increase of the PRO content. With the introduction of PRO, the elongation at break of the composite film increases significantly. Young’s modulus describes the film’s ability to resist deformation. When the PRO content is 3.96%, the young’s modulus value is up to 356.95 Pa. The addition of PRO can improve the mechanical properties of PHB films. In the stretching process, the space for the movement of the chain segment in PHB is small, and the molecular movement is very difficult to achieve. Therefore, the PHB shows obvious brittle fracture characteristics during the tensile process. After the addition of PRO, the crystal regularity and continuity of PHB are destroyed, and the crystallinity is decreased. Therefore, the toughness of the film is improved. When the PRO content reaches a certain threshold, PRO aggregates and mixes unevenly in the PHB carrier, which results in the decrease of mechanical properties of the composite films. The improvement is of great significance to the practical application of antifungal films.

#### 2.1.3. Water Solubility and Contact Angle

The water solubility of the films is shown in Figure 3. After 1 day, CK and PHB/PRO films with a loading content of 1.73% had almost no weight loss, and the weight loss rates of PHB/PRO films with loading contents of 3.96%, 8.75%, 18.15%, and 37.32% were 0.21%, 0.52%, 1.38%, and 0.91%, respectively. With the increase of time and the content of PRO in the film, the mass loss rate of PHB/PRO films in water increases gradually. After 35 days, the quality loss rates of the films with pesticide loading of 1.73%, 3.96%, 8.75%, 18.15%, and 37.32% in sterile water were 0.82%, 2.38%, 4.53%, 9.61%, and 6.74%, respectively. The mass loss was mainly due to the gradual dissolution of PRO in water. However, the loss rate of the film with 37.32% loading content was lower than that of the film with 18.15%, which may be due to the gradual dissolution of the PRO in water and the precipitation crystals of PRO attached to the surface of the film when it reached saturation. As a consequence, the quality loss was reduced. As indicated in Figure 4, the contact angle of water droplets on the film surface changed slightly within 10 min. The changes in the contact angle basically went from 80.86° to 62.60°. The data showed that all the films were hydrophobic films and there were no significant differences between the contact angles. Overall, the water solubility and contact angle tests of the composite films showed that the films have the potential for a long service life.

#### 2.1.4. SEM Analysis

The morphologies of the surface and cross section of the PHB and PHB/PRO films were determined by SEM. As shown in Figure 5, the surfaces of the CK and composite film (1.73%) were smooth and flat, which indicated that the compatibility between PRO and PHB is relatively good under a low loading content. The PRO was homogeneously distributed throughout the film. When the content of PRO is higher than 18.15%, the PRO would agglomerate into irregular shapes of different sizes and randomly disperse in the matrix, probably due to the mixture being thermodynamically immiscible [23].

SEM was also used to observe the morphology of the film after being dissolved in water for 35 days (Figure 6) and degraded in the soil for one month (Figure 7). After one month of dissolution in water, fine cracks were observed on the surface and the film began to become uneven. The crystals of PRO on the film surface decreased or even disappeared. This indicated that PRO was dissolved and released into the water. Soil was found to be the most natural environment for PHAs degradation. The PHB/PRO composite films would degrade in a soil environment. The SEM micrographs show various surface morphology changes, including alterations in the appearance of pores, cavity, grooves, and cracks. Such changes are possibly due to the existence of microorganisms in soil that secrete PHB depolymerase enzymes, which lead to the biopolymer films degradation [24].

#### 2.1.5. FT-IR Analysis

The infrared spectra of PRO, PHB, CK, and PHB/PRO composite films with different loading content are shown in Figure 8. The peak at 2978 cm^−1^ corresponds to C-H asymmetric stretching vibration, while the one at 1720 cm^−1^ with strong band absorption is attributed to the characteristic carbonyl group. PHB samples show exactly the same band absorptions to those observed for CK, indicating that the chemical stability of PHB during dissolution in the chloroform and chloroform in the films was completely volatilized. PRO exhibits characteristic absorptions at 750 and 1557 cm^−1^, which can be found in PHB/PRO composite films, indicating the successful blending of PRO with PHB. None of new peaks of composite films are observed, implying that the interaction between PHB and PRO is primarily due to physical interactions rather than any covalent bond formation.

#### 2.1.6. Optical Properties

The transparency may be affected by various factors including the film thickness and it is important to consider this parameter for mulch film [25]. As shown in Figure 9A, the light transmittance percentage of PHB/PRO films could reach about 60–80% in the range of 400–1000 nm. When the loading of PRO in the composite film reached 37.32%, the transparency decreased obviously. The greater value represents lower transparency of the film.

### 2.2. Release and Release Kinetics Studies

The release profiles of PHB/PRO composite films under three different pH values of 6.3, 7.6, and 8.8 are presented in Figure 10. All the composite films showed an initial burst release followed by a slow and sustained release. The release, however, is pH responsive. Generally, the release is faster than those under neutral and weak alkaline conditions. The accumulative release rate of the composite film with a loading content of 1.73% reached 70% under a weak acid condition after 43 h, whereas at pH 7.6 and 8.8, the corresponding release values reached approximately 49%. When the content of the composite film is higher, the accumulative release can reach 87% under an alkaline condition, and up to 79% under an acid environment.

To further investigate the release mehanism of the composite films, a classical model, known as the Korsmeyer-Peppas equation (*lnM_t_/M_∞_ = nlnt + lnk*) [26] was used to fit the release data, where *t* is the accumulative release time, while *n* and *k* is the release exponent and kinetics constant, respectively. The type of the release mechanism can be inferred from the value of *n*, when *n* < 0.45, Fick diffusion dominates drug release; when 0.45 < *n* < 0.85, the release is dominated by non-Fick diffusion. When *n* > 0.85, the drug is mainly released due to the dissolution of the drug-carrying system skeleton. For the composite film with the loading content of 1.73%, the *n* value was 0.661 and greater than 0.45, indicating non-Fick diffusion. The *n* value of all the remaining curves were less than 0.43, indicating that the release occurred as a result of Fick diffusion. Our group recently reported the trifluralin microcapsules prepared with PHB as carrier materials and the release mechanism was also Fick diffusion [27].

In actual soil application scenarios, the degradation and release of PHB/PRO composite films are more complex. The PRO will release along with the PHB degradation in soil. The biodegradation rate of PHAs depends on environmental conditions including temperature, moisture, pH, and nutrient supply. Moreover, those related to the PHAs materials themselves, such as monomer composition, crystallinity, and additives can also affect the biodegradation rate [28].

### 2.3. Antifungal Properties and Pot Experiment

As shown in Figure 11, the blank PHB film had no antifungal activity against *S. rolfsii*, while the inhibition zones were formed around the PHB/PRO films with different PRO loadings. The antifungal activity increased with the increasing content of PRO. The inhibitions of PHB/PRO films with a loading content of 3.96%, 8.75%, 18.15%, and 37.32% were 41.8%, 46.7%, 54.5%, and 80.6%, respectively. The bioactivity of the composite films was due to the release of PRO, which takes effect in the control of *S. rolfsii*. The potential application of PRO/PHB films was further evaluated by a pot experiment.

The soil for the pot experiment was from the field blocks where continuous peanut cropping occurs all year round, and where serious soil-borne diseases occur. The pot experiment showed that the growth of peanut seedlings covered with a blank PHB film was better than that of peanut seedlings without a PHB film cover (Figure 12). The degradation of PHB has a certain nutritional effect on the growth of plants [29]. With the increase of PRO/PHB antifungal film content, the root length, fresh weight, and dry weight of peanut seedlings increased slightly and then decreased. It can be concluded that the growth of peanut was promoted by inhibiting the growth of pathogenic fungi at a low concentration, and the optimal content was 3.96%. However, the high concentration of PRO might cause slight damage to peanut seedlings, so the growth of peanut seedlings was slightly adversely affected. This pot experiment clearly indicated that proper selection of application concentration of pesticide is of vital importance for crop protection. After all, PRO/PHB as mulch films have potential application prospects in the controlling of *S. rolfsii* and for promoting peanut growth.

## 3. Materials and Methods

### 3.1. Materials

Poly(3-hydroxybutyrate-co-4-hydroxybutyrate) (PHB) with 87M% of P3HB determined by ^1^H-NMR was obtained from Shandong Ecomann Technology Co. Ltd. (Zoucheng, China), and the molecular weight was described in our previous report [27]. Chemically pure chloroform from Beijing Chemical Works (Beijing, China) was used as a solvent to prepare polymer films. Prothioconazole (PRO) with the purity of 98% was kindly provided by the Sichuan Huaying Chemical Co., Ltd. (Chengdu, China). All other commercially available chemicals were used as received.

### 3.2. Preparation of PHB/PRO Composite Films

Films loaded with the fungicide were prepared as follows: PHB and PRO were mixed in different mass ratios of 1.95/0.05, 1.9/0.1, 1.8/0.2, 1.6/0.4, and 1.2/0.8. The mixtures were dissolved in 40 mL of chloroform and stirred for 2–3 h until they were completely dissolved with a final concentration of 50 mg/mL. Blending films were prepared by conventional solvent-casting technique from the PHB/PRO chloroform solution using glass petri dishes (9 cm diameter) as casting surface. The films were placed in a laminar flow cabinet at room temperature for at least 2 days until the solvents had completely evaporated. Finally, the composite films with different content of PRO were obtained.

### 3.3. Determination of Actual peSticide Loading in Different Composite Films

Briefly, 0.025 g of composite films were weighed into a 50 mL of volumetric flask, and 50 mL of methanol was added. After ultrasonic treatment for 30 min, the supernatant was filtered through a 0.22 μm membrane, and subjected to HPLC analysis. The loading content of PHB/PRO composite films was determined with Equation (1):(1)$$\mathrm{Loading}\;\mathrm{content}\;(\%)=\frac{\mathrm{Weight}\;\mathrm{of}\;\mathrm{PRO}\;\mathrm{in}\;\mathrm{composite}\;\mathrm{film}}{\mathrm{Weight}\;\mathrm{of}\;\mathrm{PHB}/\mathrm{PRO}\;\mathrm{composite}\;\mathrm{film}}\times 100.$$

HPLC analysis conditions: Venusil XBP-C_18_ reversed-phase column (5 µm × 4.6 mm × 250 mm, Bonna-Agela Technologies Inc., Tianjin, China); mobile phase, (methanol: 0.2% formic acid aqueous solution (*v*/*v*) = 80:20); flow rate, 1.0 mL/min; injection volume, 5 μL; column temperature, 25 °C; and diode array detector signals, 260 nm.

### 3.4. Characterization of Composite Films

#### 3.4.1. Film Thickness

The film thickness was measured with a precision digital micrometer (Master proof 0–25 mm, ±0.001 mm, Lilienthal, Germany) at five random locations on the film. Mean thickness values for each sample were calculated and used in tensile strength (TS) calculations.

#### 3.4.2. Mechanical Properties

Mechanical properties were measured according to the Chinese National Standard (GB/T 528-2009). The films were cut into dumbbell shaped and the testing was conducted on an automated material testing system (SANS E42.503 tensile tester, SANS, Foshan, China) at a displacement speed of 10 mm/min. The testing was performed in triplicates.

#### 3.4.3. Scanning Electron Microscopy

The surface and cross-sectional morphology of the selected film was observed with a SEM (Quanta Q400, FEI, Lacassine, LA, USA). The cross-sectional specimen was prepared by immersing the selected film into liquid nitrogen and then breaking it off. The film was fixed on the specimen holder using double-sided adhesive tapes, coated with gold, and then observed with an accelerating voltage of 2 kV. SEM was also used to observe the film after one month of dissolution in sterile water and degradation in soil. The films were cut into pieces (3 cm × 3 cm) and placed in a pot filled with soil. After one month, the sample was taken out, washed with distilled water, dried at 100 °C for 2 h, and observed using SEM.

#### 3.4.4. Water Solubility

The method for determining the water solubility rate of films was according to the literature with slight modification [30]. Approximately 0.3 g of different composite films were dried at 100 °C for 2 h and then the initial mass (M_0_) was recorded. The films were placed in 50 mL of distilled water at room temperature. After different intervals, the solution was removed and the remaining films were dried in 100 °C for 2 h. The final mass (M_t_) was recorded. The water solubility rate was calculated as Equation (2):(2)$$\mathrm{Water}\;\mathrm{solubility}\;(\%)\;=\frac{\left(M_{0}-M_{t}\right)}{M_{0}}\times 100$$

#### 3.4.5. Contact Angle

The contact angles (θ) were measured using an OCA20 video optical contact angle measuring instrument. Contact angles formed between the liquid−solid interface was determined by placing one drop of ultrapure water (4 μL) onto the film surface. Experiments were carried out at 25 °C. At least five measurements per drop were done and the resulting angles were analyzed.

#### 3.4.6. FT-IR Spectroscopy

Fourier transform infrared (FT-IR) analysis under attenuated total reflection mode was carried out to identify the functional groups present in the pristine PHB, PRO, and PHB/PRO films. An average of 30 scans with a resolution of 4 cm^−1^ was performed for all the samples over a wavelength range of 400 to 4000 cm^−1^.

#### 3.4.7. Film Transmittance and Transparency

The transmittance and transparency were determined using a UV spectrophotometer (Shimadzu, UV-1800, Tokyo, Japan). The suitable film was cut into 10 mm × 30 mm and placed on the internal side of spectrophotometer cells. The measurement was performed in the wavelength range of 200~1000 nm [18]. Three samples per treatment were tested and the transparency value at 600 nm was calculated according to Equation (3) [31]:(3)$$\mathrm{Transparency}\;\mathrm{value}\;(\%)=\frac{-\log T_{600}}{x}\times 100$$
where *T*_600_ is the transmittance percentage at 600 nm and the *x* is the thickness (mm) of the film. The greater value represents lower transparency of the film.

### 3.5. Films Release and Release Kinetics Under Different pH

Approximately 30 mg of composite films was submerged in 100 mL of release medium which was composed of phosphate buffered saline (PBS), ethanol, and Tween-80 emulsifier (70: 29.5: 0.5, *v*/*v*/*v*). The release behaviors were studied under three different pH values of 6.3, 7.6, and 8.8 on a dissolution tester (D-800LS, Tianjin University, Tianjin, China) with a stirring speed was set as 100 r/min. At the predesigned interval, 0.6 mL of release medium was taken out and an equal volume of fresh release medium was supplemented. The released amount of PRO was assayed by HPLC. The determination process was performed in triplicates. The accumulative RPO released was calculated according to the following Equation (4):(4)$$E_{r}=\frac{V_{e}\sum_{i=\;0}^{n-1}C_{i}+V_{0}C_{n}}{m_{pesticide}}\times 100\%$$
where *E_r_* is the accumulative release (%) of PRO from the composite film; *V_e_* is the volume of the release medium withdrawn at a given time interval (*V_e_* = 0.6 mL); *V_0_* is the total volume of release solution (*V_0_* = 100 mL); *C_n_* (mg/mL) is the PRO concentration in release medium at time *n*; *m_pesticide_* is the total amount of PRO embedded (mg).

### 3.6. In Vitro Antifungal Properties

The *S. rolfsii* used in this study was provided by College of Science, China Agricultural University. It was routinely cultured on potato dextrose agar (PDA) plates (90 mm) at 25 °C in the dark and then cooled to 4 °C to be used as a fungal source. The bioactivities of the films were determined by the mycelium growth rate method. The dishes without film and the film without pesticide were as controls. The dishes were incubated in the temperature-controlled cabinet at 25 °C for 72 h; then, the radius of the fungal mycelium was observed, and the degree of fungus growth inhibition relative to the controls was determined. All procedures were done in triplicate.

### 3.7. Pot Experiment

The peanut seeds of Huayu No. 23 were selected in the experiment. The seeds were surface-sterilized with 0.3% (*v/v*) sodium hypochlorite for 2 min, rinsed thoroughly with deionized water three times. After disinfection, peanut seeds were germinated by the petri dish filter paper method. After the germ broke through the seed coat for 1 cm, seeds with consistent growth were selected and transplanted to the seedling tray. Each treatment was replicated three times. The soil for seedling raising comes from the plots where continuous peanut cropping occurs all year round and soil-borne diseases occur seriously. The soil with a depth of 5–20 cm was taken back to the laboratory to remove plant residues, rocks, etc., and then the soil was thoroughly passed through a sieve with a 2 mm pore diameter. The pots were cultured in the greenhouse, and the same amount of water was poured into each pot every day to maintain proper conditions for growth. Two weeks after planting, the root length, stem length, fresh weight, and dry weight of peanut seedlings were measured.

## 4. Conclusions

This study was the first one to propose a method to control *S. rolfsii* of peanut by using natural biodegradable pesticide mulch films prepared by blending polymeric PHB with fungicide of PRO. It is also important to note that PRO with an optimal content (1.73–3.75%) has a good inhibiting effect on soil-borne diseases and a beneficial effect on peanut growth. The suitable PRO/PHB mulch films were selected through the evaluation of the mechanical properties, water solubility, and light transmittance of the films. The release of PRO from composite films was slow and sustained. The degradation of the films in the soil would accompany the growth of peanut during the whole growth period. Thus, an effective and sustained control against soil-borne diseases could be provided. When insecticides are embedded within the PHB films, the underground insects can be potentially controlled. This method would alleviate environmental pollution and human health threats caused by traditional methods. Moreover, this work indicates that a facile but powerful strategy may find wide applicability in sustainable plant and horticulture protection.

## Figures and Tables

**Figure 1 molecules-26-00762-f001:**
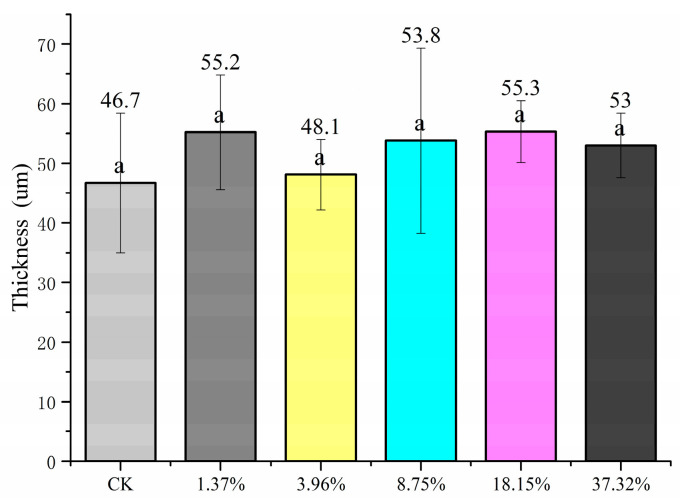
The films thickness of poly(3-hydroxybutyrate-co-4-hydroxybutyrate) (PHB) (CK) and PHB/prothioconazole (PRO) composite films. The same lowercase letters “a” in the column showed there is no significant difference at the 0.05 level when using Duncan’s new multiple range test.

**Figure 2 molecules-26-00762-f002:**
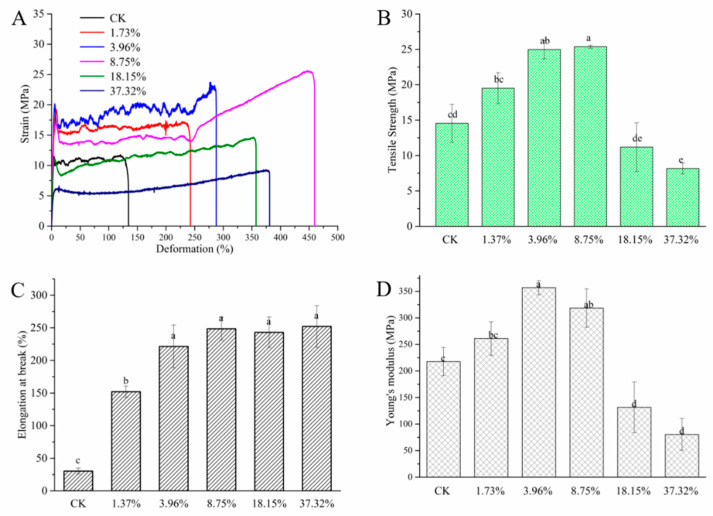
Mechanical performance of the different blending films. (**A**) stress-strain curve, (**B**) tensile strength, (**C**) elongation at break, and (**D**) Young’s modulus for of PHB and PHB/PRO blending films with different loading content. The graph represents mean value ± SE followed by the letter, the lowercase letters in the column showed there is no significant difference and different lowercase letters in the column showed significant differences in different groups at *p* ≤ 0.05.

**Figure 3 molecules-26-00762-f003:**
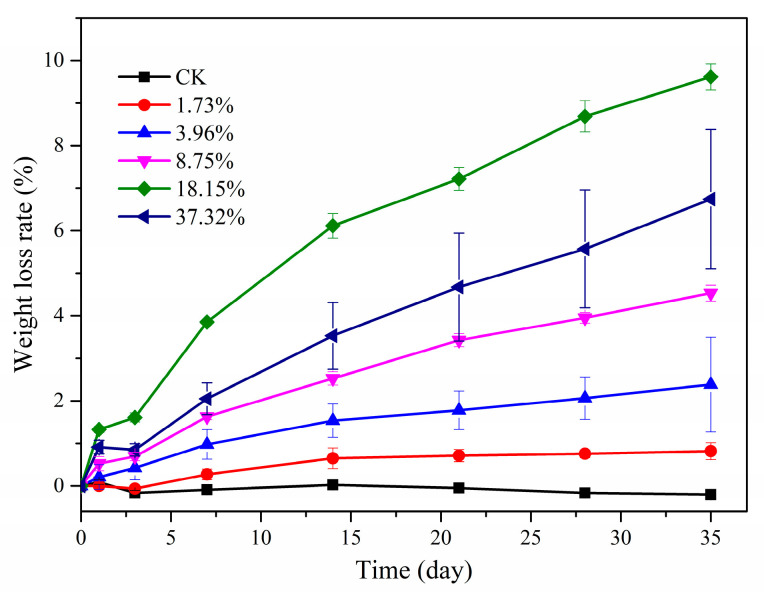
Water solubility of PHB film (CK) and PHB/PRO composite films with different loading content.

**Figure 4 molecules-26-00762-f004:**
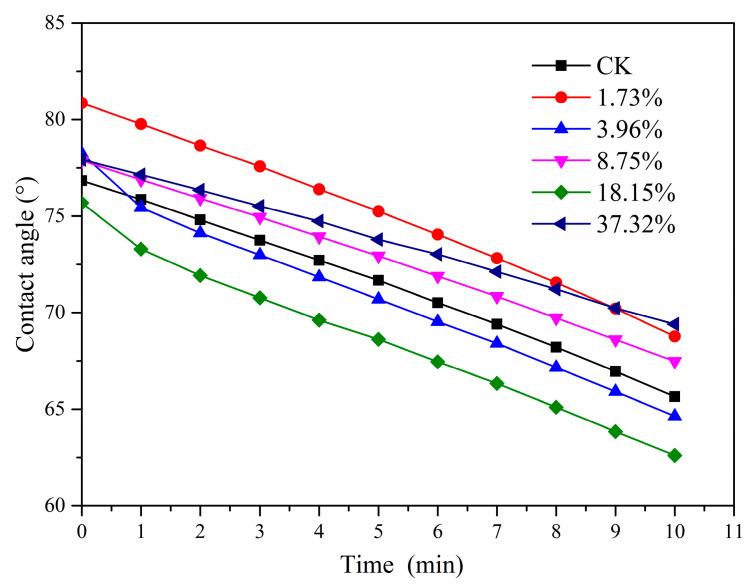
The change of contact angle on a film surface in 10 min.

**Figure 5 molecules-26-00762-f005:**
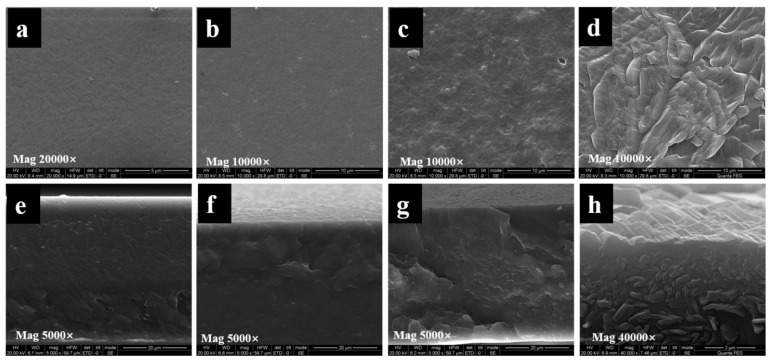
SEM images of the surfaces and cross sections of PHB film (CK, **a**,**e**) and PHB/PRO composite films with a loading content of 1.73% (**b**,**f**), 18.15% (**c**,**g**), and 37.32% (**d**,**h**).

**Figure 6 molecules-26-00762-f006:**
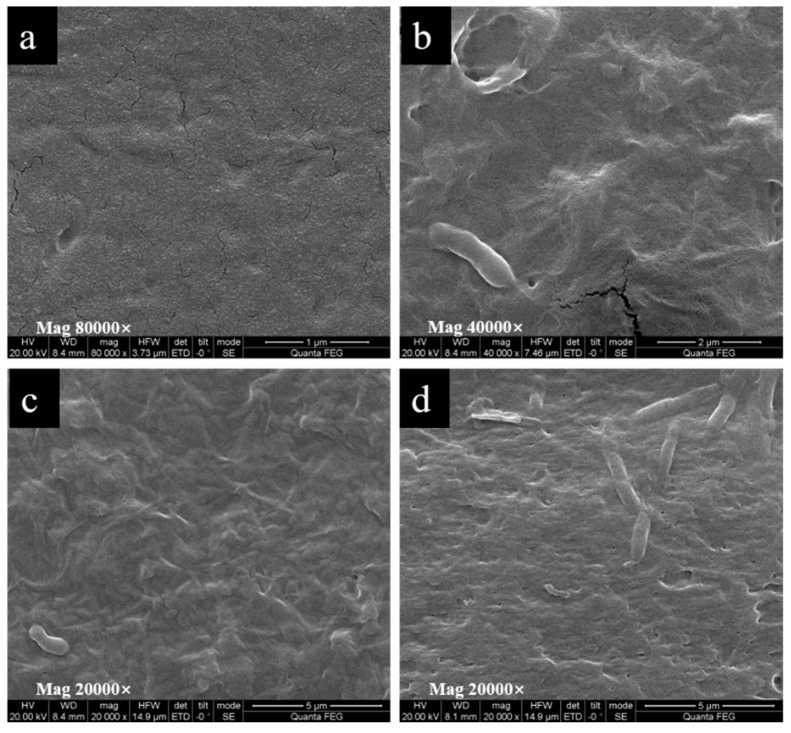
The SEM images of the composite films after dissolution in water for 35 days: (**a**) CK, (**b**) 1.73%, (**c**)18.15%, (**d**) 37.32%.

**Figure 7 molecules-26-00762-f007:**
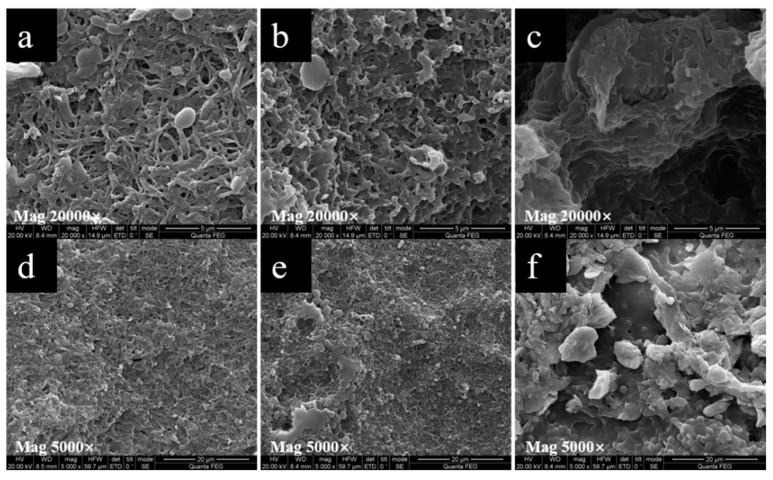
SEM images of the surfaces of PHB/PRO composite films degraded in the soil for one month: (**a**,**d**)1.73%, (**b**,**e**) 18.15%, (**c**,**f**) 37.32%.

**Figure 8 molecules-26-00762-f008:**
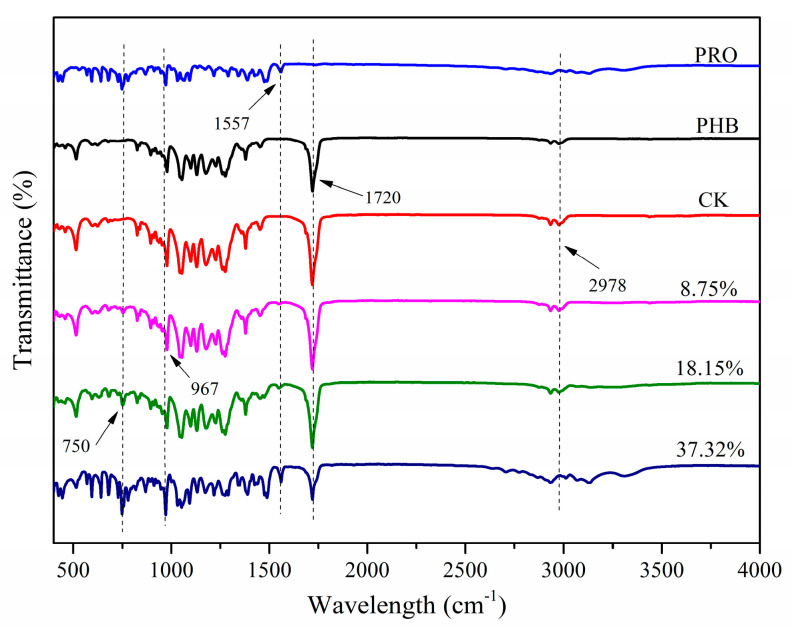
The FT-IR spectra of PRO, PHB, CK, and PHB/PRO composite films with different loading content.

**Figure 9 molecules-26-00762-f009:**
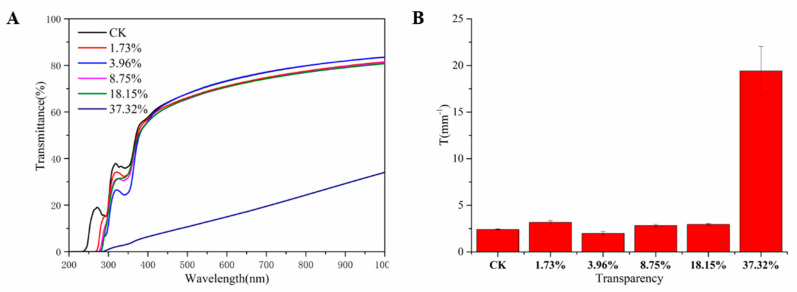
(**A**) The transmittance percentage of light different blending films and (**B**) the transparency for different blending films.

**Figure 10 molecules-26-00762-f010:**
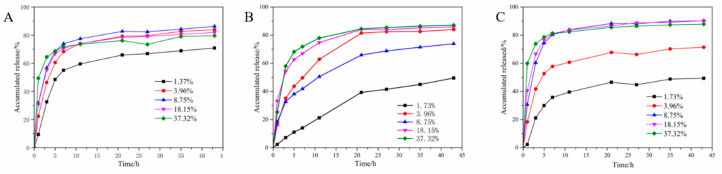
The accumulated release rates of different blending films under three pH values of 6.3 (**A**), 7.6 (**B**), and 8.8 (**C**).

**Figure 11 molecules-26-00762-f011:**
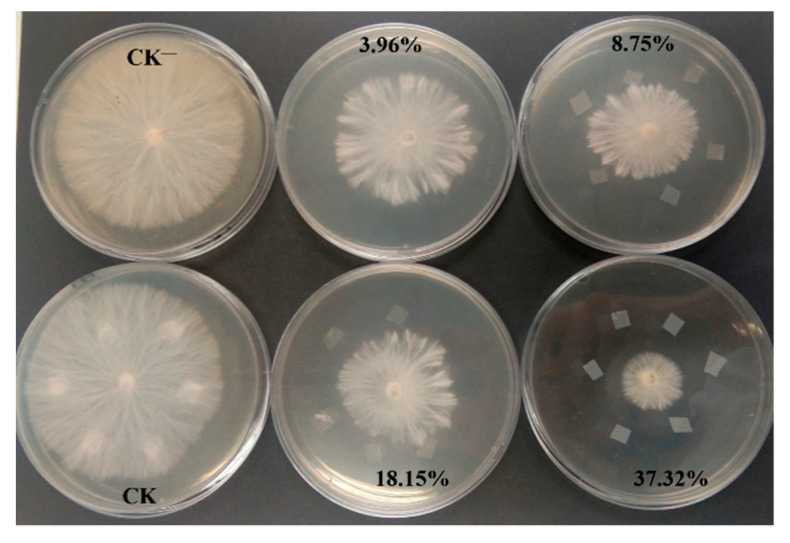
Images of the fungicidal activities of CK−(without film), CK (PHB film), and PHB/PRO composite films with different loading contents against *S. rolfsii.*

**Figure 12 molecules-26-00762-f012:**
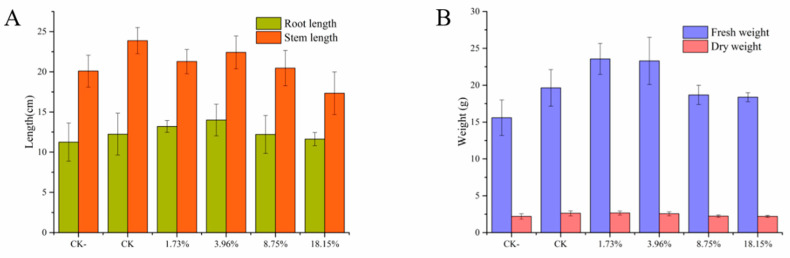
Effect of treatment on the growth of peanut in terms of root and stem length (**A**) as well as fresh and dry weight (**B**): CK−(without film), CK (PHB film), and PHB/PRO films with different loading content.

## Data Availability

The data presented in this study are available in this article.

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
