# Peer review of "Effective and Sustained Control of Soil-Borne Plant Diseases by Biodegradable Polyhydroxybutyrate Mulch Films Embedded with Fungicide of Prothioconazole"

_molecules, 2021, doi:10.3390/molecules26030762_

Round 1

Reviewer 1 Report

The authors described biodegradable antifungal mulches in the manuscript. The materials were made based on poly (3-hydroxybutyrate-co-4-hydroxybutyrate) (PHB) with fungicide of prothioconazole (PRO). Biodegradable mulch films containing the fungicide PRO have demonstrated the ability to effectively inhibit soil pathogenic fungi and can potentially be widely used in plant protection in horticulture.

I suggest a few changes:

1) Line 53-54: CO2 and H2O should be corrected.

2) In introduction, it is worth emphasizing the scientific novelty, as well as comparing the materials with other similar solutions.

3) Line 108: “The addition of PRO can improve the mechanical properties of PHB films.” Can the authors explain why the addition of PRO affects the mechanical properties of polymer compositions? Are there any reactions between PRO and PHB? Can the PRO additive affect the crystallinity of the PHB/POR blend?

4) Figure 5, 6 & 7: please add a magnification to the description of photos.

5) Do the authors analyzed the structural changes of materials after pot experiment, using FTIR spectroscopy? Can the addition of POR affect the degradation of the polymer matrix?

Author Response

The authors described biodegradable antifungal mulches in the manuscript. The materials were made based on poly (3-hydroxybutyrate-co-4-hydroxybutyrate) (PHB) with fungicide of prothioconazole (PRO). Biodegradable mulch films containing the fungicide PRO have demonstrated the ability to effectively inhibit soil pathogenic fungi and can potentially be widely used in plant protection in horticulture.

General response: First thanks very much for your positive comments on our work. It’s very kind of you to point out the deficiencies of our original manuscript. We have carefully read the comments and made revision accordingly to improve the quality of the present work. We sincerely hope this revised version can address the concerns you mentioned.

Detailed comments are as follows:

1)  Line 53-54: CO2 and H2O should be corrected.

Response: Thanks for your suggestion. In the revised manuscript, CO2 and H2O have been corrected.

2)  In Introduction, it is worth emphasizing the scientific novelty, as well as comparing the materials with other similar solutions.

Response: Thanks for your suggestion. In the revised manuscript, we have added the contents of the Introduction. Specific additions are as follows: In the existing application of PHB-supported pesticide films, the researches on the mechanical and optical properties of the films are rarely reported. Other biodegradable materials that integrated herbicides with mulching film for sustained and effective weed control have been reported [16-17]. In the application of biodegradable plastic mulching film for controlling soil-borne disease, Liang et al. prepared biodegradable water-triggered chitosan/hydroxypropylmethyl cellulose metalaxyl mulch film for sustained control of Phytophthora sojae in soybean [18].

3)Line 108: “The addition of PRO can improve the mechanical properties of PHB films.” Can the authors explain why the addition of PRO affects the mechanical properties of polymer compositions? 

Response: The addition of PRO can enhance the toughness of the composite film. With the increase of PRO content, the tensile strength of the composite antibacterial film increases firstly and then decreases, and the elongation at break increases gradually. In the stretching process, the space for the movement of chain segment in PHB is small, and the molecular movement is very difficult. Therefore, it shows obvious brittle fracture characteristics during the tensile process. After the addition of PRO, the crystal regularity and continuity of PHB are destroyed, and the crystallinity is decreased. Therefore, the toughness of the film is improved. When PRO content reaches a certain threshold, PRO aggregates and mixes unevenly in PHB carrier, which results in the decrease of mechanical properties of the composite films.

Are there any reactions between PRO and PHB?

Response: The infrared spectra of PRO, PHB, CK, and PHB/PRO composite films with different loading content are shown in Fig. 8. The results shows that none of new peaks of composite films are observed, implying that the interaction between PHB and PRO is primarily due to physical interactions rather than any covalent bond formation.

Can the PRO additive affect the crystallinity of the PHB/POR blend? 

Response: The PRO additive can affect the crystallinity of the PHB/PRO blend. After the addition of PRO, the crystal regularity and continuity of PHB are destroyed, and the crystallinity is decreased. As a result, the toughness of the film is improved.

4) Figures 5, 6 & 7: please add a magnification to the description of photos.

Response: Thanks for your suggestion. We have added magnification in the picture description.

5) Do the authors analyzed the structural changes of materials after pot experiment, using FTIR spectroscopy?

Response: Thanks for your suggestion. Infrared spectroscopy is a common and rapid method to test the chemical functional groups of materials. It should be very interesting to measure the structural changes of films after pot experiment by using infrared spectroscopy, which will guide our research in the future. In the present study, the structural and surface changes of the films degraded in the soil for one month were clearly observed by SEM images.

Can the addition of PRO affect the degradation of the polymer matrix?

Response: According to the SEM images results, when the content of PRO is low, the change of surface morphology of the composite film after degradation for one month is not significant compared with that of PHB film. When the content of PRO reaches 37.32%, the morphology changes significantly. Based on these results, it can be concluded that the addition of PRO affects the degradation of the polymer matrix. However, the influence was not studied in-depth in the present work. The degradation of composite film in soil and its effect on soil microorganism will be the focus of future research.

Reviewer 2 Report

Dear Authors

In the present work, biodegradable antifungal mulch was prepared by mixing poly (3-i-16 droxybutyrate-co-4-hydroxybutyrate) (PHB) with protioconazole (PRO) fungicide, which were used for effective and long-lasting soil control. borne plant diseases. To reveal the application perspectives of PHB / PRO composite films in the management of soil-borne plant diseases, some physical and biological properties were evaluated. The appropriate mulch film of PHB / PRO was evaluated based on mechanical and optical properties, water solubility, and the micromorphology of the film was further characterized. The release patterns of composite films with different pHs were investigated. Furthermore, the in vitro antifungal biological assay and the pot experiment showed satisfactory bioactivity of the  PHB / PRO films against Sclerotium rolfsii Sacc., A soil-borne disease in peanut fields. This study has shown that biodegradable mulch films containing PRO fungicide are able to effectively inhibit plant pathogenic fungi present in the soil and this easy but powerful strategy can find wide applicability in the sustainable protection of plants and vegetables. I believe that the paper was written comprehensibly and also the clear methods and the results well explained.

Author Response

Thanks very much for your positive comments on our work. We have checked our manuscript carefully and made some revision accordingly to improve the quality of the present work.

Round 2

Reviewer 1 Report

I recommend the publication of this article in the Molecules journal.